# Current Opinion in LAIV: A Matter of Parent Virus Choice

**DOI:** 10.3390/ijms23126815

**Published:** 2022-06-19

**Authors:** Irina Kiseleva

**Affiliations:** Institute of Experimental Medicine, 197376 St. Petersburg, Russia; irina.v.kiseleva@mail.ru

**Keywords:** influenza vaccine, classical reassortment, reassortant, wild-type parent virus, virus properties, live attenuated influenza vaccine

## Abstract

Influenza is still a frequent seasonal infection of the upper respiratory tract, which may have deadly consequences, especially for the elderly. This is in spite of the availability of vaccines suggested for persons above 65 years of age. Two types of conventional influenza vaccines are currently licensed for use—live attenuated and inactivated vaccines. Depending on local regulatory requirements, live attenuated vaccines are produced by the reverse genetics technique or by classical reassortment in embryonated chicken eggs. Sometimes, the efficiency of classical reassortment is complicated by certain properties of the wild-type parent virus. Cases of low efficacy of vaccines have been noted, which, among other reasons, may be associated with suboptimal properties of the wild-type parent virus that are not considered when recommendations for influenza vaccine composition are made. Unfortunately, knowledge surrounding the roles of properties of the circulating influenza virus and its impact on the efficacy of the reassortment process, vaccination efficiency, the infectivity of the vaccine candidates, etc., is now scattered in different publications. This review summarizes the main features of the influenza virus that may dramatically affect different aspects of the preparation of egg-derived live attenuated vaccine candidates and their effectiveness. The author expresses her personal view, which may not coincide with the opinion of other experts in the field of influenza vaccines.

## 1. Introduction

Influenza virus is responsible for 3–5 million cases of severe illness worldwide and up to 650 thousand respiratory deaths [1]. These figures speak for themselves and provide strong evidence that effective influenza vaccines are needed by mankind. The key process underlying annually recurrent epidemics is the influenza virus’s rapid evolution [2,3,4,5,6,7]. Punctuated antigenic changes lead to escape from the immunity [8] that was previously induced by infection or vaccination. Humankind eradicated smallpox and plague, but we are still like children in the face of attacks of influenza virus that continue to cause death after more than 100 years since its first appearance. Since those times, tremendous progress has been made in studying influenza. “The paradox of the current situation is a clear contradiction between the achievements of theoretical science and practical results in the fight against influenza infection. Science has created all theoretical prerequisites for defeating influenza. Nevertheless, the study on influenza and influenza viruses is still at its beginning…” [9] and vaccination remains the main tool of influenza control. Flu viruses are constantly changing; therefore, unlike many other viral vaccines, which keep the same composition over time, the formulation of influenza vaccines is revised every influenza season.

Currently, there are two major types of traditional vaccines licensed for use—live attenuated (LAIV) and inactivated (IIV), which include whole inactivated virus vaccines, subunit vaccines, split virus vaccines, and virosomal influenza vaccines [10]. Every year, vaccine manufacturers distribute nearly 500 million doses of IIV and LAIV worldwide [11]. People vaccinated by injection of IIV can acquire the influenza infection, but the disease is less severe in those who have been vaccinated. In contrast, LAIV is administered intranasally and directly protects the entry gate of infection; thus, the risk of infection is significantly lower.

Current influenza vaccine candidates, for potential use in vaccine manufacturing, are reassortants of master donor virus (MDV) with wild-type (WT) virus that is antigenically similar to the recommended strain. MDVs have all the necessary characteristics for the type of vaccines of which they are intended. Two types of MDVs are used in the preparation of influenza vaccines—high-yielding donors for IIV and temperature-sensitive (*ts*) and cold-adapted (*ca*) donors of attenuation—for LAIV.

One of the key advantages of LAIV over IIV is that, unlike IIV, it induces humoral, mucosal, and cellular immune responses [12,13,14,15]. LAIVs may require less complex downstream processing, which would be more appropriate for technology transfer. In addition, cell-culture-derived production technology for LAIV is under development.

LAIVs have a lower unit cost and higher production yield, estimated to be 10-times higher than for inactivated vaccines [13]. These qualities are particularly important when preparing for an influenza pandemic [13].

The variety of influenza virus subtypes and their constant variability have turned the search vector towards universal vaccines, which are most often based on short conservative fragments of hemagglutinin (HA) or M-protein, enclosed in platforms of a different nature. The variety of platforms is considerably greater than the many conservative sequences used as antigens [10]. These are all experimental and not yet licensed vaccines, although I–III phases of the clinical trial have been completed for a number of universal vaccines [16]. There is no shortage of variety in their combinations, experimented on for over a decade. Today, however, a true universal influenza vaccine remains elusive. The development of new vaccines has not yet been crowned with real success—in terms of effectiveness, they are still somewhat less effective than the existing vaccines licensed for use [10,17]. Therefore, improving the effectiveness of already existing licensed vaccines is of great importance.

Since the beginning of the COVID-19 pandemic, the circulation of influenza viruses in the world has dramatically decreased. However, this has not eliminated the need for the regular preparation of influenza vaccines. There are certain reasons for this. Vaccination, even in conditions of limited circulation of influenza viruses, can reduce the severity of mixed "COVID-19 + influenza" infections and protect the most vulnerable people [18,19]. In addition, surprisingly, a rapid increase in global H3N2 influenza virus activity has been noticed since the end of 2021 [20]. Therefore, influenza vaccination remains relevant.

Speaking of protective measures against influenza, we should not forget about simple hygiene protective equipment, such as masks. The World Health Organization (WHO) recommends the use of medical masks to fight influenza pandemics and as a precautionary measure during the coronavirus pandemic. Today, in the era of COVID–19, we have begun to forget about such a dangerous infection as the seasonal flu, and in vain. The WHO, in its recommendations for the prevention of seasonal influenza, provides several steps, the main of which is vaccination, but does not mention such simple but effective measures as wearing masks [21]. Nevertheless, this measure can reduce the risk of disease in families, in communities, and especially, in hospitals [22], among patients with aggravated anamnesis, persons from risk groups, etc. If vaccines are not effective enough or they are not available at all, individual hygiene and protection measures are put forward in the first place. Unfortunately, at the beginning of the recent pandemic, humanity made many irreparable mistakes. Perhaps, in the case of a wider use of masks, many fatal outcomes could have been avoided.

The recommendations on the composition of influenza virus vaccines are general and do not differentiate between vaccine types, live or inactivated. This review describes the biological properties of the WT virus that would affect the efficiency and properties of the vaccine reassortment but are not taken into account when giving recommendations on the composition of influenza vaccines. 

A large body of literature describing the insufficient efficacy of influenza vaccines exists [23,24]. This may be due to the mismatch between the circulating and vaccine strains, especially when it comes to an inactivated vaccine. It is also possible that the not-satisfactory quality of some influenza vaccines is precisely due to the biological properties of the parental viruses that are described below in this review article. The quality and effectiveness of any influenza vaccine is largely determined by the right choice of the parental virus. If developers of influenza vaccines are guided not only by the antigenic novelty of the strain recommended by the WHO, but also carefully select, among similar viruses, one that, in addition to antigenic novelty, has all the necessary biological properties, the effectiveness of influenza vaccines may increase.

The question remains, how do we choose the best WT parent? To address the challenges of choosing the best WT virus for LAIV, this article reviews certain important biological properties of WT influenza viruses that may help or prevent a vaccine strain to be the best vaccine candidate. The points made are considered from the standpoint of specialists who regularly prepare LAIV candidates by the method of classical reassortment.

## 2. The World Health Organization Recommendations for the Composition of Influenza Vaccines

In 1973, the WHO made the first annual recommendations for the influenza vaccine composition [25]. As of today, every February, the WHO recommendations for the Northern Hemisphere are issued, and every September, for the Southern Hemisphere, respectively [26]. 

Occasionally, the recommendations have been a surprise [26], for example, in February 2020. That is why influenza vaccine manufacturers, pending recommendations, prepare dozens of strains based on recently circulating viruses, eventually finalizing only one. This is very costly, both in terms of resources and time involved. 

Since the 1980s, two genetic lineages of influenza B virus—B/Victoria and B/Yamagata—have been circulating among the human population, with one or the other lineage being more prevalent in specific regions of the world [27,28]. In 1999, the problem of choosing recommendations between different strains of influenza B virus first arose. In previous years, when the trivalent vaccine was used, once for the Northern Hemisphere (in 1999–2000) and once for the Southern Hemisphere (in 2000), two B viruses were simultaneously recommended (B/Shangdong/7/97-like virus of B/Victoria lineage and the B/Beijing/184/93-like virus of B/Yamagata lineage). Decisions as to which one was the most appropriate B component were made by national control authorities based on local epidemiological data [26].

Until 2012, seasonal vaccines included three strains of influenza viruses (trivalent vaccine contained A(H1N1), A(H3N2), and B/Yamagata or B/Victoria lineage viruses). On February 23, 2012, for those considering the use of quadrivalent vaccines containing two influenza B viruses of B/Victoria and B/Yamagata lineages, viruses of both lineages were first recommended for the 2012–2013 influenza season [26]. For the Southern Hemisphere, the first recommendations for a quadrivaccine were made on September 20, 2012 (2013 influenza season).

Unfortunately, due to the constant emergence of antigenically new strains, the effectiveness of the strains recommended by the WHO is not guaranteed; half of the forecasts for 1999 to 2021 were concluded as not optimal. For instance, during the 1997–1998 season, a considerable mismatch between the A(H3N2) vaccine component and the most prevalent epidemic influenza A(H3N2) virus was noticed [29]. In February 2019, the WHO Influenza Committee found it difficult to recommend a vaccine strain of the A(H3N2) subtype on time and the recommendation for the A(H3N2) component was postponed for a month—A(H3N2) virus was announced on March 21, 2019.

In the late 1980s, synthetic chemist Kary Mullis discovered a method for amplifying DNA using the polymerase chain reaction (PCR) [30]. Between the discovery of PCR and the present day, there was a 35-year-long gap. Today, sequencing has become a routine method, but in the 1990s–2000s, it was not widely used in the analysis of the genome composition of vaccine candidates or in the selection of WT parent viruses. Their choice was traditionally based on the assessment of the virus prevalence in the circulation and serological reactions. The main emphasis was on exceeding the four-fold difference in antibody titers in the HI test [31]. 

As mentioned, the WHO issues recommendations for the respective hemispheres on an annual basis on the eve of a new influenza season, but this does not mean that every strain in influenza vaccines is updated every year (Figure 1, Figure 2 and Figure 3). 

In the past, the vaccine strains were not always changed annually. In more recent times, the WHO more often and more regularly began to change the strains and the vector of linear progression, shifting towards their annual replacement. However, looking at the trend of replacement of influenza A(H1N1) (Figure 2) and especially A(H3N2) vaccine virus strains (Figure 1), it is unlikely that this is caused by the increasing speed at which influenza viruses evolve. Rather, more accurate methods of assessment of virus novelty have emerged. 

Obviously, changing strains in vaccine composition follow the changing antigenic structure of influenza viruses. It is known that the fastest evolving viruses are A(H3N2), then A(H1N1), and the slowest is influenza B [8]. From 1997 to the latest recommendations of February 25, 2022, for the 2022–2023 influenza season in the Northern Hemisphere, the A(H3N2) vaccine component was changed 17 times for 26 seasons (Figure 1). 

During the same time interval, H1N1 viruses changed in vaccines only nine times (Figure 2), B/Yamagata—seven times; B/Victoria—seven times (Figure 3).

As mentioned, until 2012, only trivalent vaccines were produced and, accordingly, only one strain of influenza B virus was recommended—either B/Victoria or B/Yamagata lineages. It is not possible to build an adequate linear prognosis for each influenza B lineage due to the relatively small number of recommended viruses belonging to each lineage (Figure 3). A very similar picture can be observed for the Southern Hemisphere (Appendix A).

Since the late 1970s, a number of the WHO position papers on influenza and influenza vaccines and several WHO requirements for LAIVs and IIVs have been published [12,32,33,34,35]. These requirements for LAIV and IIV were revised several times; they described—in detail—the manufacturing and control process for vaccine preparation. However, clear rules were never established for what properties a WT virus should have, other than its antigenic novelty and relevance. The current factors that are officially used for virus selection for the seasonal influenza vaccine are as follows: (i) surveillance data indicate which viruses are currently circulating; (ii) antigenic characterization monitors for changes in currently circulating viruses; and (iii) serology tests provide information about the similarity of currently circulating viruses and viruses included in vaccines for the previous influenza season [36].

From the author’s point of view, today’s approach for making recommendations is somewhat mechanistic, since it does not consider the variety of key biological features of the virus. Today, there is a pronounced tendency towards molecular genetic analysis, with the antigenic novelty at the molecular level taken into account. However, the pathogen itself, the pathogenesis, and the biological properties of the virus, which are often critical in the preparation of the vaccine strain, are forgotten. There is no doubt that the fine molecular structure and antigenic novelty of HA and neuraminidase (NA) are important, but the approach to strain selection should not be so mechanistic. It seems reasonable that new potential candidates for vaccine strains should be evaluated not only in terms of the novelty of surface antigens but also by taking into account changes in the biological characteristics of viruses and their pathogenetic effects on the host.

The recommendations on the composition of influenza virus vaccines do not differentiate between vaccine types, although the biological properties of the vaccine candidates are fundamentally different for live and inactivated vaccines. IIV requires high growth reassortants of the highest titer of HA; infectivity of these reassortants is not as important. On the contrary, for LAIV, the infectious activity of the vaccine virus is of importance, while its hemagglutinating activity does not play a key role. Therefore, it may be necessary to make separate recommendations for IIVs and LAIVs. In this context, it is worth noting that IIV only protects against the strain it contains. In contrast, LAIV provides a wider spectrum of immune response and protection compared with IIV [14]. Antigenic mismatch between circulating and incorrectly recommended strains, which may cause low vaccine effectiveness, is less important for LAIV than for IIV. Perhaps LAIV composition does not need to be changed frequently to ensure it remains effective and we should consider not recommending new LAIV strains as often as for IIV strains.

## 3. Derivation of Conventional Reassortant Influenza Vaccines; Optimal Genome Composition of the Vaccine Candidate

Four cold-adapted (*ca*, adapted to replication at low temperatures of 25–26 °C) viruses are currently in use as donors for licensed LAIVs—A/Leningrad/134/17/57 (H2N2), A/Ann Arbor/6/60ca (H2N2), B/USSR/60/69, and B/Ann Arbor/1/66ca. LAIV candidates on the backbone of A/Leningrad/134/17/57 (H2N2) and B/USSR/60/69 MDVs are made by classical reassortment in embryonated chicken eggs (further simply referred to as “eggs”). Briefly, eggs are inoculated with a mixture of WT virus recommended by the WHO for vaccine development and MDV. The progeny of the co-infection is subjected to a number of selective passages and clonings under two selective conditions—temperature of 25–26 °C and the presence of anti-MDV serum [37,38]. About six–seven passages are performed after co-infection of parental viruses (two selective passages, one–two blind passages, if necessary, two–three clonings, and one amplification) [37,38,39,40,41].

The mandatory genome composition of the LAIV candidate is 6:2 (six genome segments from the *ca* MDV, and HA and NA from the seasonal circulating strain) [37,38,39,41]. LAIV candidates based on A/Ann Arbor/6/60ca (H2N2) and B/Ann Arbor/1/66ca are RG-generated, with the same 6:2 genome composition as the classical vaccine [14,42,43]. Six internal genes of *ca* MDVs transfer to LAIV candidate temperature sensitivity (*ts*, an inability for replication at the temperatures above optimal range), *ca* phenotype, and the ability to actively replicate in eggs at the optimum temperature [37,44,45].

Reassortants for whole-virion conventional IIV are derived by the pressure of only one selective factor—antiserum against donor of internal genes [46]. After co-infection, the virus progeny is subjected to five–nine post-reassortment passages with or without serum. Vaccine candidate derivation reports are available in [46]. Genome composition requirements for this type of influenza vaccine are more flexible. The parental source for the internal genes of IIV reassortants differs with different manufacturers. Genome composition of IIV candidates is also varied (6:2, 5:3, 1:7, 1:1:6, 1:2:5, etc.). 

The main condition is that the genome composition of the IIV candidate should contain gene coding for HA of the WT seasonal virus and include those internal genes from the donor strain that ensure its high yield. For instance, the M gene of A/PR/8/34 (H1N1) (PR8) is particularly required for a high yield of influenza A IIV candidates [47,48,49]. The NP gene of B/Lee/40 contributes to the improved growth of influenza B reassortants for IIV [50,51]. Reassortants for IIV are derived by classical reassortment. Both conventional viruses (PR8, A/Texas/1/77 (H3N2), B/Lee/40, etc.) and hybrid strains, for instance, 5:3 reassortant IVR-6 (A/Texas/1/77 × PR8), 4:4 reassortant NYMC BX-46 (B/Lee/40 × B/Panama/45/90), etc., are used as high-yielding donors. A full list of IIV candidate viruses, their genome composition, and parent strains can be found on the The National Institute for Biological Standards and Control (NIBSC) website [46].

For classical reassortment in eggs, nature creates optimal viable combinations; typically, reassortants of 6:2 genome compositions can be obtained with a greater or lesser degree of difficulty. However, sometimes at “natural co-infection” desired genome composition 6:2 is not optimal and other, more viable combinations may prevail. Certain genome compositions might have reduced viability, indicative of their functional incompatibility. For instance, Subarrao et al. [52] were not able to reassort human A(H1N1) viruses with gull influenza A(H13N6) viruses as a donor of internal genes with the desired 6:2 genome composition. Classical reassortment failed to generate 6:2 LAIV candidates based on H5N2 or H5N1 parents and the backbone of A/Leningrad/134/17/57 (H2N2); only 7:1 gene constellation was achieved [45,53,54]. 

A(H5N2) [55] or A(H1N1)pdm09 [56,57] 5:3 conventionally derived reassortants were shown to produce higher yields than their 6:2 counterpart. Gilbertson et al. [58] demonstrated that 5:3 vaccine candidates containing a PB1 gene segment from A(H1N1)pdm09 virus and HA and NA from A(H5N1) and A(H7N9) viruses provided higher yields, suggesting that a particular growth advantage is conferred to reassortant by the PB1 gene of A(H1N1)pdm09. 

In 2009, to enhance their replicating properties, two reassortant viruses for the commercial IIV (X-181 and X-181A) were developed, using conventional reassortment technology using the 5:3 gene constellation with three genes (PB1, HA, and NA) obtained from A/California/07/2009 (H1N1)pdm09 and the remaining five genes originated from A/PR/8/34 [56,57]. Constellation 6:2 was possibly inefficient for the vaccine strain to acquire the best properties. As such, 5:3 LAIV reassortant (HA, NA, and M genes originated from WT parent) was more immunogenic for mice than the 6:2 candidate in the HI test [59].

One reason why the classical reassortment technique for LAIV production may be preferred over reverse genetics (RG) approaches is because it allows for the natural selection of a variant with an optimal gene constellation that improves growth properties in the reassortant in eggs [49,60]. The question, “which genome composition of LAIV candidate is better?” exists—would it be the genetically engineered 6:2 reassortant of dubious quality or the highly productive 7:1 naturally obtained reassortant? 

Of course, 6:2 can be forcefully engineered using the RG technique, but nature does not tolerate the disruption. Would such a 6:2 reassortant be good enough? Is it necessary to go against nature if the optimal constellation is not 6:2, but, for instance, 5:3 or 7:1? There is evidence that conventionally derived 7:1 pandemic live attenuated influenza vaccines are safe and immunologically effective in clinical trials [61,62].

## 4. Naturally Occurring Temperature-Sensitive WT Influenza Viruses

An important characteristic of any virus is its *non-ts* phenotype—an ability for replication at the elevated temperatures of 38–40 °C, which exceed the upper limit of optimal values. In the past, it was thought that the typical WT virus is always *non-ts* and that this property determines viral virulence. In those times, primary screening of reassortant LAIV candidates was based on *ts/ca* attenuation markers [63,64]. Reassortants that did not contain suitable laboratory markers of attenuation (*ts/ca* phenotype) were screened out; only then, analyses of the genome composition, which at that time was quite complex, started. The *ts*-phenotype of the reassortant LAIV candidate is critical because *ts* viruses cannot multiply at the temperature of the lower respiratory tract.

The first mention of natural *ts* WT influenza viruses can be found in publications from the 1980s [65,66,67,68,69]. Later, it was found that changes in the *ts/non-ts* phenotype have a regular wave-like nature [70,71,72]. At the beginning of each influenza pandemic/epidemic cycle, the circulation of *non-ts* viruses was detected. Further evolution is leading to the change in *non-ts* with *ts* variants. The prevalence of *ts* strains in circulation indirectly indicates that novel *non-ts* viruses are expected to appear in circulation. 

Thus, the permanent circulation of *ts* viruses can be considered as a precursor for the appearance of an antigenically distinct virus, seasonal or even pandemic. This assumption is supported by the fact that just before the 2009 pandemic, a kind of “calm before the storm” was noticed: only *ts* influenza A(H1N1), A(H3N2), and B viruses were detected in circulation [70].

The *ts* phenotype of the WT virus does not influence the efficiency of reassortment but interferes with the efficacy of primary screening of the egg-derived LAIV candidate, since one of the laboratory selective markers of attenuation, the *ts* marker, is lost. The problem of the existence of *ts* viruses lies elsewhere—since *ts* viruses are at the end of the *ts* wave of circulating viruses, they may be less immunogenic than their *non-ts* counterparts, whose circulation started this wave. It has been suggested that *non-ts* WT parent viruses may enhance the immunogenicity of LAIV and vice versa; LAIVs based on *ts* WT parent viruses were of low immunogenicity. Unfortunately, this study only tested a limited number of vaccines [72].

It seems reasonable that new potential candidates for vaccine strains should be evaluated not only in terms of the novelty of surface antigens but also taking into account temperature sensitivity in their replication.

## 5. Naturally Occurring Cold-Adapted WT Influenza Viruses

In nature, not only natural temperature-sensitive viruses circulate, which are numerous, but sometimes *ca* viruses also appear, which usually possess the *non-ts* phenotype [73]. Unlike natural *ts* viruses, there are so few that it is not possible to make any assumptions about the reasons and regularities in their appearance. In fact, the term “*ca*” (cold-adapted) is not quite appropriate in this case, since these viruses were not adapted to low temperatures by laboratory manipulations. It would be more accurate to talk about WT viruses that sufficiently replicate at low temperatures or WT viruses that are naturally resistant to low temperatures. The role of cold-resistance for replication of some natural isolates has not been studied yet; however, it can be assumed that *non-ts/ca* WT viruses, which can reproduce in a very wide temperature range, from 25 °C to 40 °C, can effectively infect both the upper and lower respiratory tract.

Unlike *ts* WT viruses whose *ts* phenotype does not influence the efficiency of reassortment, the *ca* phenotype of the WT parent dramatically disturbs the first steps in the reassortment process that are carried out at a low temperature of 25–26 °C. A loss of the *ca*-selective factor may lead to a significant increase in the total number of *ca* reassortants, but the overall number of reassortants with the desired 6:2 genome composition is decreasing.

## 6. Sensitivity of WT Viruses to Nonspecific Thermostable Serum γ-Inhibitors

WT influenza viruses exhibit marked differences in their sensitivity to nonspecific thermostable γ-inhibitors due to their distinguishable receptor specificity. H3N2 viruses, which preferentially bind the α-2,6 receptors, are very sensitive to serum thermostable γ-inhibitors, while H1N1 strains with α-2,3 or mixed α-2,3/α-2,6 specificity exhibit an inhibitor-resistant phenotype [74,75,76]. Before the 1970s–1980s, the majority of influenza B viruses possessed an inhibitor-resistant phenotype. In the 1980s, they diverged into two distinct genetic lineages, B/Victoria and B/Yamagata [28]. Since then, there have been two parallel evolutionary pathways of influenza type B in the human population [27]. After separation, the B/Victoria lineage viruses retained a high level of inhibitor resistance of past strains. Contrarily, viruses of the B/Yamagata lineage acquired high inhibitor sensitivity [77].

The standard scheme for the preparation of vaccine strains by the method of classical reassortment includes the use of anti-MDV serum [37,38,41]. This provides a selective advantage for reassortants in inheriting HA and NA from an antigenically relevant WT virus. However, the selection of 6:2 reassortants, based on inhibitor-susceptible WT viruses, can be complicated by nonspecific binding of their HA by γ-inhibitors, which are presented in the anti-serum against MDV [76]. 

The data presented in [76] were used for drawing Figure 4. Analysis of genome composition of 883 reassortants, obtained by classical reassortment in eggs of MDVs with 40 WT influenza viruses, which possessed a different degree of sensitivity to nonspecific γ-inhibitors, revealed the following consistent pattern: all reassortants inherited WT HA; nevertheless, the belonging of the remaining genes to WT or MDV parents varied [76]. The majority of reassortants based on inhibitor-resistant WT viruses (88.7%) inherited NA from the WT parent; also, the highest percentage of 6:2 reassortants (31.4%) was achieved (Figure 4, left panel). 

In contrast, the efficiency of obtaining 6:2 reassortants was much lower (7.2%) if the WT parent virus possessed a high degree of sensitivity to nonspecific thermostable γ-inhibitors; clones with the 7:1 genotype (25%) prevailed among the obtained reassortants. Corruption of the constellation of genes encoding HA and NA was observed—only a quarter of all reassortants inherited both WT HA and WT NA and three-quarters had WT HA + MDV NA [76]) (Figure 4, right panel). 

Thus, the inhibitor sensitivity of WT viruses becomes an obstacle to the effective preparation of vaccine reassortants for LAIV by classical reassortment, since immune serum against MDV is involved in the selection process/screening of vaccine candidates. Contrarily, the inhibitor resistance guarantees a faster and more stable result in the preparation of vaccine strains. The development of LAIV based on the classical reassortment method would benefit from the recommendation of viruses with a high level of resistance to inhibitors. On the other hand, inhibitor-sensitive viruses retain a preference for α-2,6-linked residues. For now, the question, “what should be the best vaccine strain—inhibitor-sensitive or inhibitor-resistant?”, remains open.

## 7. Infectivity of WT Viruses and LAIV Candidates

One of the key indicators of the quality of reassortant candidates for IIV is their high HA titer. There has been up to a 512-fold increase in HA titers of PR8-based vaccine reassortant observed as compared to the respective WT parent virus [49]. Reassortants that produce high HA titers do not always have a high yield of infectious viruses [78] but infectious viral titers of reassortant candidates are not so critical for IIV. For example, reassortants prepared on a high-yielding PR8 donor, NIBRG-23 (H5N1) and VN/PR/CDC-RG (H5N1), displayed rather low infectious viral titers, which did not exceed 6.2 log_10_ EID_50_/mL and 7.7 log_10_ EID_50_/mL, correspondingly [40,45].

On the contrary, infectivity is critical for LAIV. Whereas the WT parent virus typically has relatively low infectious titers (6.2–7.7 log_10_ EID_50_/mL [45]), the titers of LAIV candidates on the backbone of *ca* MDV are usually 8.7–10.2 log_10_ EID_50_/mL [37,40,45]. 

As for the viruses to be recommended, typically, a reference strain and a few reference strain-like viruses that are similar in antigenic properties to the reference virus are recommended. Sometimes reference strain-like viruses appear to be less or more effective in the development of reassortant vaccine candidates than reference strains. For instance, based on our experience, the reassortant LAIV candidate of A/Leningrad/134/17/57 MDV with A/Brisbane/34/2018 (H3N2) WT parent (A/Kansas/14/2017-like virus recommended for use in 2019-2020 Northern Hemisphere influenza season) displayed ~ 1 log_10_ EID_50_/mL higher infectious activity than the reassortant candidate based on the A/Kansas/14/2017 (H3N2) reference strain, respectively. In contrast, the reassortant LAIV candidate of A/Leningrad/134/17/57 MDV with the A/Michigan/173/2020 (H3N2) WT parent (A/Darwin/9/2021-like virus recommended for use in 2022-2023 Northern Hemisphere influenza season) displayed ~ 1.0–1.5 lg_10_ EID_50_/mL lower infectious activity than the reassortant candidate based on the A/Darwin/9/2021 (H3N2) reference strain, respectively (I. Kiseleva, E. Bazhenova, E. Stepanova, N. Larionova and L. Rudenko. Personal communications).

Interestingly, the reassortment of *ca* MDV with PR8-based vaccine strains for IIV of relatively low infection titers led to a dramatic increase in infectivity of the resulting reassortants [40,45]. Unfortunately, there are cases when the presence of genes from an attenuated ca MDV does not significantly increase the infectious viral titers of reassortants. This has been observed in recent years for A(H3N2) influenza viruses and may be related to the receptor specificity of these viruses. If A(H1N1) and A(H1N1)pdm09 influenza viruses possess α-2,3 or α-2,3/α-2,6 specificity, due to which they multiply well in eggs without prior adaptation, then A(H3N2) influenza viruses retaining a preference for α-2,6 specificity [74] have always been a problem for reproduction in eggs, becoming even more serious recently. Sometimes, national influenza centers that conduct year-round surveillance for influenza were not able to isolate A(H3N2) viruses in eggs to be recommended for seasonal vaccines in a timely manner.

## 8. Thermal and pH Stability of the HA of Influenza Viruses

Membrane fusion activity and infectivity of influenza virus require cleavage of HA0 [79]. Some studies indicate that low pH and high temperature could trigger membrane fusion activity and co-vary with changes in HA conformation [80,81,82]. A decrease in pH after the virus enters the endosome causes an irreversible change in the HA conformation, which is necessary for the fusion of the outer layer of the virus with the endosome membrane: this is a way that the virus enters the cytoplasm. A change in the conformation of HA can also occur when exposed to high temperatures. It was demonstrated that the pH threshold, when HA is losing its stability, varies. Among influenza viruses isolated in the 1960s–1980s, H3N2 viruses were found to be relatively stable against low pH (threshold between 5.1 and 5.4). H1N1 viruses were intermediate in this respect. Most of the avian influenza viruses that possessed H5HA and H7HA were relatively labile (pH threshold 5.6–6.0) [83]. In contrast, the HA of more recent highly pathogenic H5N1 avian influenza viruses was shown to be more susceptible to low pH than that of human viruses [84].

Based on the results obtained with the H5N1 avian influenza virus, which has low immunogenicity for humans, Krenn et al. [84] suggested that the sensitivity of H5HA to the acidic environment might compromise the immunogenicity of intranasal influenza vaccine and cause its low infectivity. Wolkerstorfer et al. [59] also found that increased pH of HA activation may lead to decreased virus infectivity and immunogenicity of the LAIV. It is very likely that vaccine effectiveness may require a certain level of pH and temperature stability of HA in the vaccine virus to induce a sufficient immune response. 

A similar situation was observed with H1N1pdm09 strains. A number of studies demonstrated poor protection of LAIV based on H1N1pdm09 pandemic viruses in 2013–2016 [85]. In the USA, these data even led to an extraordinary decision to not recommend LAIV for use in the 2016–2017 influenza season [86]. Destabilizing mutations that affect viral resistance to high temperature and low pH appeared in the HA of 2009 influenza pandemic viruses. Vaccine candidates based on these had low thermal stability in HA, were sensitive to low pH, had low immunogenicity, and had low stability for fluctuations in ambient temperature [87]. The HAs of the pandemic strains isolated after 2010 were more stable. The improvement in their stability was attributed to a novel Glu-47-Lis substitution in the HA2 subunit of the stalk region. This single amino acid substitution affects viral fusion, pH, thermal stability, and infectivity [87,88].

There is one more point regarding the low A(H1N1)pdm09 LAIV effectiveness—it was explained by the presence of defective interfering (DI) RNAs. High amounts of DI viral RNAs in vaccine preparation may have contributed to the low effectiveness of LAIV [89,90]. However, the mechanisms of this phenomenon are not entirely clear. The problem of arising DI RNAs may be rooted in certain properties of the influenza virus, for instance, sensitivity to interferon, or pH optimum of HA-mediated membrane fusion [90,91]. 

Even within the same subclade, certain properties in viruses can vary significantly. For instance, among 21 H1N1pdm09 viruses, regardless of year of isolation, 9 viruses possessed high thermal stability in the HA and 12 viruses were of low thermal stability [92]. Therefore, from a number of viruses, the possibility to select the one that will have the desired properties always exist.

## 9. Pros and Cons of Reverse Genetics

The discovery of the polymerase chain reaction [30] and plasmid-based RG technology [93] allows the rapid generation of reassortant vaccine candidates [94,95]. The RG approach has a number of undeniable advantages in the preparation of vaccine strains. The RG technology allows artificial manipulation with the influenza virus genome and is a powerful tool to generate a surrogate reassortant virus with any desired genomic composition in a short period of time. In addition, the RG approach reduces potential antigenic changes—when rescuing genetically engineered reassortants, the same sequence of the viral genome that was initially laid down is obtained, although, when amplifying in eggs, additional mutations may occur [57,93,95,96].

As for pandemics and potential pandemic influenza vaccines, it should be remembered that the availability of influenza vaccines in these situations is largely dependent on the vaccine virus yield. During a pandemic, when large amounts of vaccines have to be produced quickly, the RG method may be preferred over the classical approach. Moreover, only the RG method allows producing candidate reassortant vaccines for highly pathogenic avian influenza viruses with the deletion of determinants of high pathogenicity in HA (polybasic cleavage site) [57,97].

MedImmune LAIV candidates are currently produced by reverse genetics [5,67,68,69] (Figure 5, approach 1). For other countries, using the RG approach to prepare vaccine viruses is restricted by the necessity to purchase a license from the patent holders. In Russia, genetic manipulations with LAIV candidates are officially prohibited; reassortants for Russian LAIVs are being made by classical genetic reassortment in eggs [37,40] (Figure 5, approach 2). Regardless of the method used for preparation, LAIV candidates retain a complete set of attenuating mutations and, as a consequence, maintain the *ts/ca* phenotypes that are typical for the MDVs.

During the reassortment performed by classical co-infection of the WT virus with MDV, the virus progeny passes at least seven times in eggs (co-infection of parent viruses, two selective passages, one–two blind passages if necessary, two–three clonings, and one amplification) [37,38,39,40,41]. Additional egg passages, which are made during a process of the manufacturing of LAIV, should also be considered. As a result, the reassortant acquires additional egg-adapted mutations. It leads to the predominant selection of high-growth reassortants efficiently multiplied in eggs. It is as if during the process of the microevolution of virus in eggs, the most adapted to replication in eggs and most viable clones are naturally selected. This is especially critical for the recent A(H3N2) viruses. According to our experience, genetically engineered A(H3N2) reassortants for LAIV often lag behind reassortants produced by classical reassortment in terms of growth characteristics in eggs.

Thus, theoretically, the RG method is quicker, since there is no need to spend time on the first stages of classical reassortment. In fact, the quality of the RG method may be unsatisfactory; artificial reassortants may be less viable and we may need to additionally pass virus progeny through eggs several times to achieve additional adaptation and high infectious virus titers [105].

## 10. Egg-Adapted Mutations in Hemagglutinin; Cell-Derived Vaccines 

Recently, there have been several publications that describe molecular changes in the HA associated with the adaptation of the influenza A virus to replication in eggs. Current H3N2 influenza viruses possess a glycosylation site that alters the binding of antibodies elicited by egg-adapted vaccine strains and thereby, may alter the antigenicity and dramatically decrease vaccine effectiveness [106,107,108,109,110,111,112,113]. Most attempts to reselect an influenza virus possessing sequence changes in HA characteristics of mammalian cells, by multiple passages in MDCK cells of the egg-derived isolate, were not successful [114]. Thus, retaining the original HA sequences of clinical isolates during vaccine production might be crucial for the LAIV effectiveness [115].

The fact that circulating strains isolated in mammalian cell culture are closer to clinical specimens but are distinguishable antigenically from their egg-derived counterparts has been known for a long time [116,117]. The situation became even more complicated when it appeared that for some influenza virus strains, eggs were not as efficient for primary isolation of human influenza viruses as cell cultures were [118]. This created a big problem for vaccine manufacturers because the majority of influenza vaccines are produced in eggs. For instance, the fiasco in the isolation of the A/Fujian/411/2002 (H3N2) virus in eggs resulted in its absence in the vaccine for the 2003–2004 season. Consequently, the A/Moscow/10/99 (H3N2)-like virus was recommended for the fourth influenza season in a row. Only by the next season, egg-adapted the A/Fujian/411/2002 (H3N2)-like virus, was selected for vaccine preparation (Figure 1) [111]. This could be due to the fact that human-cell-derived H3N2 isolates have been reported to bind with a high affinity to α2-6-linked sialosides, while viruses isolated in eggs have often increased specificity for α2-3-linked sialosides [119,120,121].

The limited availability of egg isolates, particularly of recent H3N2 viruses, which grow poorly in eggs, may lead to serious problems in the selection process of vaccine virus candidates. Therefore, timely recommendations for the use of H3N2 egg-derived influenza vaccine candidates could be delayed.

How can these problems be avoided? Cell-culture-derived influenza vaccines seem like a good solution. Since 2018, representatives of the predominant circulating human influenza viruses have been recommended by the WHO for vaccine production, both in eggs and in cell lines [26]. The choice of substrate for vaccine manufacturing depends entirely on the current regulatory requirements of specific countries [122].

Cell-culture-derived influenza vaccines, compared to egg-based technology, reduce production time and the risk of contamination. They are safe for those with an allergy and animal-component-free production is achievable. Cell-derived vaccines are devoid of egg-adaptive changes.

Two strategies can be utilized for the development of cell-based LAIVs: (i) development of reassortants are made in eggs and final stages (amplification on an industrial scale) are performed in cells [100,101,102,103] (Figure 5, approach 3) or (ii) all stages of the vaccine preparation starting from WT virus isolation are made exclusively in cells [104,105] (Figure 5, approaches 4 and 5). The first strategy possesses a significant disadvantage—the resulting reassortant vaccine candidate will contain egg-adapted mutations, which may alter the vaccine antigenicity compared to its human population circulating counterparts. The second strategy is more logical because vaccine candidates will be free of egg-adapted mutations.

The use of cell culture will allow vaccine manufacturers to become independent of the egg supply. An experimental Vero-derived vaccine obtained by a classical genetic reassortment was described [105]. All stages of the vaccine development were performed in Vero cells, including the isolation of the WT viruses. The authors did not mention problems associated with reassortment in cell culture but the Vero-derived H1N1 vaccine candidate did not provoke a measurable antibody response in healthy volunteers [105]. In contrast, conventionally derived reassortants generated in MDCK cells sometimes possessed the majority of genes of the WT parent; researchers encountered unpredictable difficulties in the regular development of the 6:2 reassortant candidates [123]. However, despite the difficulties of preparation of MDCK-derived LAIV, it was shown to be safe and immunogenic for healthy volunteers [104].

All these problems with undesirable egg-adapted mutations and difficulties of reassortment in cell culture can be solved by reverse genetics. A cell-culture-derived LAIV, based on cell culture, isolates and develops using reverse genetics techniques (Figure 5, approach 4) that may have a great future compared to its antigenicity undistinguished from clinical specimen counterparts. Such vaccines may demonstrate increased effectiveness compared to conventional egg-derived vaccines. A similar approach has already been described in the literature for IIV [124]. In the future, cell-derived genetically engineered LAIV based on cell-derived viruses, possessing all critical biological features, may replace the conventional time-consuming, labor-intensive egg-based influenza vaccine production technology.

## 11. Are Viruses Alive? Pro et Contra

All of the above leads to the eternal question—are viruses alive? If not, then what are they? Scientists are still not sure whether viruses are living or non-living [125,126,127,128,129,130,131,132,133] so the answer remains unclear. Nonetheless, for 10 “pro” publications [125,126,127,128,129,132,134,135,136,137], on average, there is 1 “contra” [130]. 

If viruses are not living things, they are just complicated assemblies of organic molecules (nucleic acids, proteins, lipids, etc.) that are not able to multiply until they enter a living cell. If the virus is only a complex organic molecule, then the only thing that needs to be done, to announce new recommendations of influenza vaccine composition, is to track subtle changes in its chemical/antigenic structure, which is now being done by the WHO. 

On the other hand, an organism can be considered as alive if it can be infected with something. When virophages—new virus species that are parasites of other viruses—were discovered [134,135,136,137], this suggested that viruses could be alive. If so, they have a number of certain biological properties that affect both the pathogenesis of the infection they cause and the characteristics of the vaccine strains developed on their basis. Therefore, one cannot be limited to an exclusively mechanistic approach in the process of making a recommendation on influenza vaccine composition.

## 12. Conclusions

It is possible that in the future, when/if universal influenza vaccines are finally licensed, the issue of the biological properties of WT viruses and their antigenic novelty will not be so acute, and recommendations on influenza vaccine composition will be made in a different vein. Nevertheless, until this happens, it is necessary to think about how to make recommendations more flexible. Which WT strain should be recommended for the influenza vaccine from a vaccine developer’s point of view? The author and associates believe that to develop highly effective LAIVs, in addition to antigenic relevance and novelty, certain key biological properties of the chosen WT parent virus that may jeopardize the development of LAIV candidates should also be considered. A combination of the *non-ts* phenotype, stability of HA to heating and low pH, the ability to actively reproduce in an appropriate substrate, etc., is the key to successful and timely preparation of LAIV candidates. Then, the recommended strain with the optimal combination of these features will serve as the ideal basis for a highly immunogenic LAIV candidate of high infectious viral titers. Within the viruses of similar genetic subclades/lineage circulating within the same area and at the same time, their individual properties can vary significantly. Therefore, the possibility to select the one that will have the desired properties from a number of antigenically homogeneous viruses always exists. Of course, selecting a strain based on its extended characteristics will take a little longer for recommending authorities, but “the game is worth the candle.”

It should be emphasized that the author expresses her personal view, which may not coincide with the opinion of other experts in the field of influenza vaccines.

## Figures and Tables

**Figure 1 ijms-23-06815-f001:**
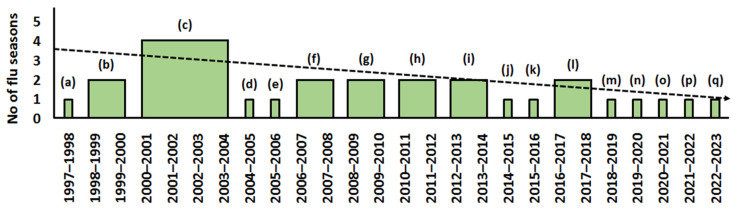
Recommended influenza A(H3N2) virus vaccines for use in Northern Hemisphere influenza seasons (based on [26]). A(H3N2) strains: (a) A/Wuhan/359/95-like virus; (b) A/Sydney/5/97-like virus; (c) A/Moscow/10/99-like virus; (d) A/Fujian/411/2002-like virus; (e) A/California/7/2004-like virus; (f) A/Wisconsin/67/2005-like virus; (g) A/Brisbane/10/2007-like virus; (h) A/Perth/16/2009-like virus; (i) A/Victoria/361/2011-like virus; (j) A/Texas/50/2012-like virus; (k) A/Switzerland/9715293/2013-like virus; (l) A/Hong Kong/4801/2014-like virus; (m) A/Singapore/INFIMH–16–0019/2016-like virus; (n) A/Kansas/14/2017-like virus; (o) A/Hong Kong/2671/2019-like virus; (p) A/Cambodia/e0826360/2020-like virus; (q) A/Darwin/9/2021-like virus. Dotted line—linear forecast. Axis X: Northern Hemisphere influenza seasons.

**Figure 2 ijms-23-06815-f002:**
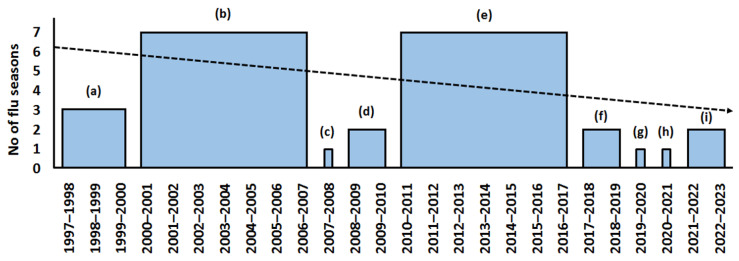
Recommended influenza A(H1N1) virus vaccines for use in Northern Hemisphere influenza seasons (based on [26]). A(H1N1) strains: (a) A/Beijing/262/95-like virus; (b) A/New Caledonia/20/99-like virus; (c) A/Solomon Islands/3/2006-like virus; (d) A/Brisbane/59/2007-like virus. A(H1N1)pdm09 strains: (e) A/California/7/2009-like virus; (f) A/Michigan/45/2015-like virus; (g) A/Brisbane/02/2018-like virus; (h) A/Guangdong–Maonan/SWL1536/2019-like virus; (i) A/Victoria/2570/2019-like virus. Dotted line—linear forecast. Axis X: Northern Hemisphere influenza seasons.

**Figure 3 ijms-23-06815-f003:**
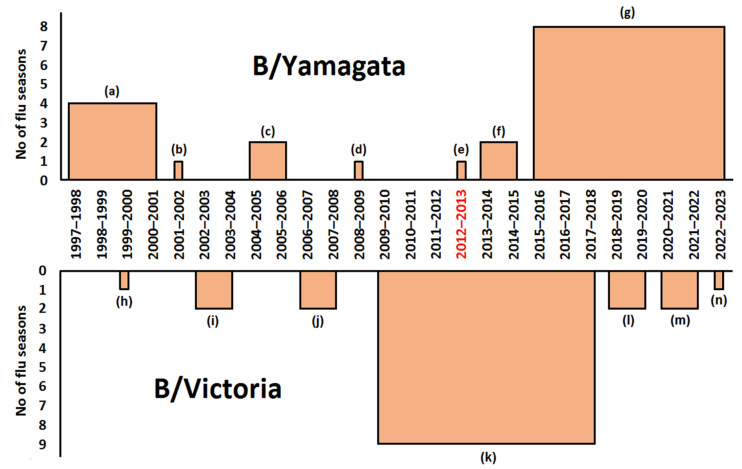
Recommended influenza B virus vaccines for use in Northern Hemisphere influenza seasons (based on [26]). B/Yamagata strains: (a) B/Beijing/184/93-like virus; (b) B/Sichuan/379/99-like virus; (c) B/Shanghai/361/2002-like virus; (d) B/Florida/4/2006-like virus; (e) B/Wisconsin/1/2010-like virus; (f) B/Massachusetts/2/2012-like virus; (g) B/Phuket/3073/2013-like virus. B/Victoria strains: (h) B/Shangdong/7/97-like virus; (i) B/Hong Kong/330/2001-like virus; (j) B/Malaysia/2506/2004-like virus; (k) B/Brisbane/60/2008-like virus; (l) B/Colorado/06/2017-like virus; (m) B/Washington/02/2019-like virus; (n) B/Austria/1359417/2021-like virus. 2012–2013 influenza season (highlighted in red) when the first WHO recommendations for the composition of a quadrivalent influenza vaccine were made. Axis X: Northern Hemisphere influenza seasons.

**Figure 4 ijms-23-06815-f004:**
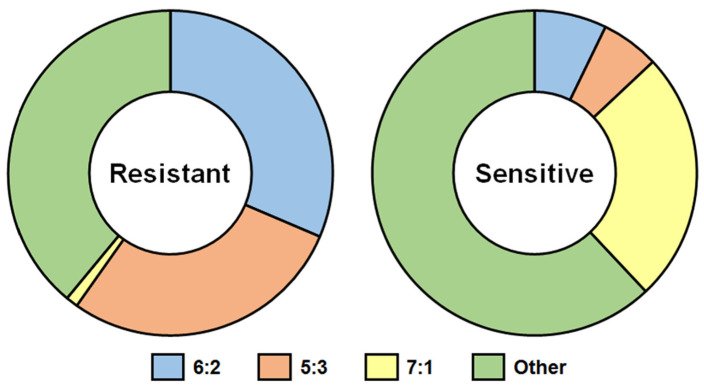
Genome composition (%) of reassortants derived by classical reassortment of *ca* MDVs with resistant or sensitive to nonspecific thermostable γ-inhibitors A(H1N1), A(H2N2), A(H3N2), A(H5N1), B/Victoria lineage, and B/Yamagata lineage WT influenza viruses [76]). Left panel—328 reassortants of *ca* MDVs with 20 WT viruses that are resistant to nonspecific thermostable γ-inhibitors were analyzed; 555 reassortants of *ca* MDVs with 20 WT viruses that are sensitive to nonspecific thermostable γ-inhibitors were analyzed. 6:2 genome composition—HA and NA are inherited from the WT parent, and 6 internal genes are inherited from MDV; 5:3 genome composition—HA, NA and one of the internal genes are inherited from the WT parent, the other five internal genes are inherited from MDV; 7:1 genome composition—HA is inherited from WT parent, all internal genes and NA are inherited from MDV.

**Figure 5 ijms-23-06815-f005:**
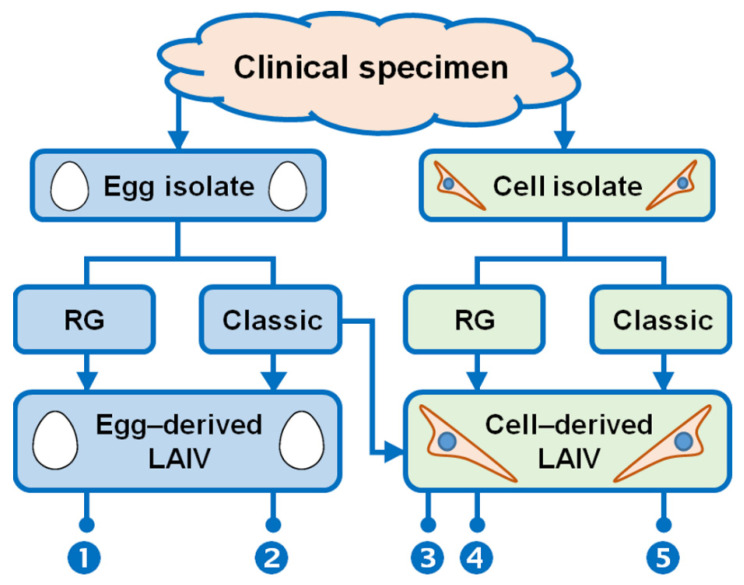
Approaches for the development of reassortant LAIV. RG—development of LAIV candidate by reverse genetics; Classic—development of LAIV candidate by classical reassortment. Approach 1—all stages of LAIV preparation are performed in eggs except RG manipulations [14,42,43,98]. Approach 2—all stages of LAIV preparation are performed in eggs [37,38,39,40,41]; Approach 3—preparation of vaccine reassortant is performed in eggs but the final stages of LAIV manufacturing are performed in cells [99,100,101,102]. Approach 4—all stages of LAIV preparation are performed in cells including RG manipulations. Approach 5—all stages of LAIV preparation are performed in cells [103,104]. Approaches 1 and 2—conventional licensed LAIVs, Approaches 3–5—experimental LAIVs for research use.

## Data Availability

Not applicable.

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
