# Peer review of "Current Opinion in LAIV: A Matter of Parent Virus Choice"

_ijms, 2022, doi:10.3390/ijms23126815_

Round 1

Reviewer 1 Report

This is a comprehensive and outstanding review and personal perspective on influenza viruses, and their vaccines.

Specific Comments for consideration are: Line 14, Suggest removing "a". Line 25. Suggest "---measure used today to control"---. Line 28. "-- drift, and the influenza -- ". Line 37.  "---with wild type (WT) -- ". Line 51. "--of hemagglutinin (HA) --- ". Line 56. Unfathomable is awkward here. Suggest "-- a true universal influenza vaccine remains elusive."  Line 67. Delete "very". Line 70. "-- but are not taken into account -- ". Lines 72-73. "-- LAIV, this article reviews ---". Lines 74-75. Delete "in this review article". Line 83.Delete "for many people". Line 84. Needs period after February 2020. Line 86. Suggest  "-- eventually finalize only one. This is  -----   and time involved." Line 89. Insert comma after "In 1999, --".  Line 105.  "--WHO is not --". Line 120. Insert comma after aforementioned -- ". Line 172. "From the author's point of view, --". Line 178. Spell out NA here, then abbreviate. Line 234. HA and WT abbreviations have been defined earlier. Line 243. Spell out NIBSC, then abbreviate. Line 267.  Delete "to" . Line 268. Spell out reverse genetics, then abbreviate as RG. Line 273. "RG".  Line 274. Suggest "disruption" rather than "violence".  Line 415 Suggest "--isolate the A(H3N2) viruses in eggs --".  Lines 461-462.  Use RG as it's already abbreviated.  Line 524. "-- for some influenza virus strains, eggs were -- ". Line 557. Suggest "--culture will allow vaccine --- to become independent -- ". Line 589. "--can be infected when something --". Line 590. "-- new virus species that are parasites of other --". Line 602. "The author and associates believe ---". Line 614. "-- will take a little longer for the recommending ---".Line 616 "---expresses her personal view, --".

Author Response

Response to the Reviewer #1 Comments

COMMENTS AND SUGGESTIONS FOR AUTHORS

This is a comprehensive and outstanding review and personal perspective on influenza viruses, and their vaccines.

Author’s response: Dear Sir or Madame, The author sincerely thanks you for all your critiques and suggestions. Your valuable comments help the author to improve the content of the manuscript. I greatly appreciate the Reviewer's positive assessment of our work and tremendous work done to improve the English language of the manuscript. Please, find answers to the Reviewer's comments below.

SPECIFIC COMMENTS FOR CONSIDERATION ARE:

Point 1: Line 14, Suggest removing “a”.

Author’s response 1: The author thanks the Reviewer for the suggestion. Indefinite article “a” was removed.

Point 2: Line 25. Suggest "–– measure used today to control" ––.

Author’s response 2: The author thanks the Reviewer for the suggestion. “Today” was added.

Point 3: Line 28. "–– drift, and the influenza –– ".

Author’s response 3: The author thanks the Reviewer for the suggestion. “And” was added.

Point 4: Line 37. "–– with wild type (WT) –– ".

Author’s response 4: The author thanks the Reviewer for the criticism. The abbreviation “WT” was defined.

Point 5: Line 51. "–– of hemagglutinin (HA) –– ".

Author’s response 5: The author thanks the Reviewer for the criticism. The abbreviation “HA” was defined.

Point 6: Line 56. Unfathomable is awkward here. Suggest "–– a true universal influenza vaccine remains elusive."

Author’s response 6: The author thanks the Reviewer for the suggestion. The sentence was changed according to the Reviewer suggestion.

Point 7: Line 67. Delete "very".

Author’s response 7: The author thanks the Reviewer for the criticism. “Very” was deleted.

Point 8: Line 70. "–– but are not taken into account –– ".

Author’s response 8: The author thanks the Reviewer for the criticism. “…but do not take into account” was replaced with “…but are not taken into account”.

Point 9: Lines 72–73. "–– LAIV, this article reviews ––".

Author’s response 9: The author thanks the Reviewer for the criticism. “…LAIV, below we will discuss certain important biological properties…” was replaced with “…LAIV, this article reviews certain important biological properties…”

Point 10: Lines 74–75. Delete "in this review article".

Author’s response 10: The author thanks the Reviewer for the criticism. “…in this review article…” was deleted.

Point 11: Line 83.Delete "for many people".

Author’s response 11: The author thanks the Reviewer for the criticism. “…for many people…” was deleted.

Point 12: Line 84. Needs period after February 2020.

Author’s response 12: The author thanks the Reviewer for the comment. Comma after “2020’ was replaced with dot.

Point 13: Line 86. Suggest "–– eventually finalize only one. This is ––––– and time involved."

Author’s response 13: The author thanks the Reviewer for the suggestions. Suggested changes gratefully accepted.

Point 14: Line 89. Insert comma after "In 1999, ––".

Author’s response 14: The author thanks the Reviewer for the comment. Comma was inserted.

Point 15: Line 105. "–– WHO is not ––".

Author’s response 15: The author thanks the Reviewer for the criticism. “…are…” was replaced with “…is…”

Point 16: Line 120. Insert comma after aforementioned –– ".

Author’s response 16: The author thanks the Reviewer for the comment. Comma was inserted.

Point 17: Line 172. "From the author's point of view, ––".

Author’s response 17: “From our point of view,” was replaced with “From the author's point of view,”

Point 18: Line 178. Spell out NA here, then abbreviate.

Author’s response 18: The author thanks the Reviewer for the criticism. The abbreviation “NA” was defined.

Point 19: Line 234. HA and WT abbreviations have been defined earlier.

Author’s response 19: The author thanks the Reviewer for the criticism. Definitions of HA and WT were removed.

Point 20: Line 243. Spell out NIBSC, then abbreviate.

Author’s response 20: The author thanks the Reviewer for the criticism. The abbreviation “NIBSC” was defined.

Point 21: Line 267. Delete "to".

Author’s response 21: The author thanks the Reviewer for the criticism. “To” was deleted.

Point 22: Line 268. Spell out reverse genetics, then abbreviate as RG.

Author’s response 22: The author thanks the Reviewer for the criticism. The abbreviation “RG” was defined.

Point 23: Line 273. "RG".

Author’s response 23: The author thanks the Reviewer for the criticism. “Reverse genetics” was replaced with “RG”.

Point 24: Line 274. Suggest "disruption" rather than "violence".

Author’s response 24: The author thanks the Reviewer for the suggestion. “…violence” was replaced with “…disruption”

Point 25: Line 415 Suggest "–– isolate the A(H3N2) viruses in eggs ––".

Author’s response 25: The author thanks the Reviewer for the suggestion. Suggested changes gratefully accepted.

Point 26: Lines 461–462. Use RG as it's already abbreviated.

Author’s response 26: The author thanks the Reviewer for the comment. Definition of “RG” was removed.

Point 27: Line 524. "–– for some influenza virus strains, eggs were –– ".

Author’s response 27: The author thanks the Reviewer for the criticism. “…for some influenza viruses, strains eggs were…” was replaced with “…for some influenza virus strains, eggs were…”

Point 28: Line 557. Suggest "–– culture will allow vaccine ––– to become independent –– ".

Author’s response 28: The author thanks the Reviewer for the suggestion. Suggested changes gratefully accepted.

Point 29: Line 589. "–– can be infected when something ––".

Author’s response 29: An organism is alive if it can be infected with something. The author meant, “An organism can be considered as alive if it can be infected with something.”

Point 30: Line 590. "–– new virus species that are parasites of other ––".

Author’s response 30: The author thanks the Reviewer for the suggestion. Suggested changes gratefully accepted.

Point 31: Line 602. "The author and associates believe ––".

Author’s response 31: The author thanks the Reviewer for the suggestion. “We believe…” was replaced with “The author and associates believe…”

Point 32: Line 614. "–– will take a little longer for the recommending ––".

Author’s response 32: The author thanks the Reviewer for the suggestion. “…will take for recommending authorities a little longer,” was replaced with “…will take a little longer for the recommending authorities,”

Point 33: Line 616 "–– expresses her personal view, ––".

Author’s response 33: The author thanks the Reviewer for the criticism. “His” was replaced with “her”

Reviewer 2 Report

This is a comprehhensive review of the efforts made to improve the efficacy of

vaccines to prevent Influenza infection,disease and death.It makes clear

that the central problem is the high antigenic mutation tendency of the different influenza strains (mainly H1N1 influenza A).

Comments:

The first sentence of the abstract is not understandable as vaccinated people can acquire the infection and the disease.Infected people can became severely ill and die in the same way as it happens for vaccinated COVID-19-patients.

Several recent publicationsreporting abbout hospitalzed severely illl influenza infected patients do not even mention the vaccination status (Kao et al 2018 Ann Intensive Care, Hussain M et la 2020 Pneumonia).Other found(retrospectively) that vaccinated persons are less often sick after influenza infection but nosocomial infections were also serious in vaccinated patients (Ye LT et al. Int J Env research and Public health 2019;16:1078,Murillo-Zamora E et al.Int J Infectious Dis 2021;112:288-293).

Chow EJ et al. found that the number of influenza positive patientes admitted to the hospital because of respiratory problems who were vaccinated was similar to that non-vaccinated. (JAMANetwork Open 2020 March 30).

It is necessary to make clear that the virus continue to cause death after more than 100 year after its first apparence.

The virus does  not reach the blood stream.How should the antibody work to avoid further worsening of the clinical conditions (e.g.hypoalbuminemia,kidney insufficiency etc.)

These are crucial questions.
Authors should also say that the best preventive measure in the hospitals is wearing masks (Ambrosch A et al.J Hospital Infection 2016;94:143-149).

Therefore, the first reccomendation (also from the WHO) should be that hospital personel should always wear masks.

that vaccinated patients

Author Response

Response to Reviewer #2 Comments

COMMENTS AND SUGGESTIONS FOR AUTHORS

This is a comprehensive review of the efforts made to improve the efficacy of vaccines to prevent Influenza infection, disease and death. It makes clear that the central problem is the high antigenic mutation tendency of the different influenza strains (mainly H1N1 influenza A).

Author’s response: Dear Sir or Madame, The author sincerely thanks you for all your critiques, suggestions and points of interest for discussion. I believe that your valuable comments help to improve the content of the manuscript. Please, find the answers to your comments below.

COMMENTS:

Point 1: The first sentence of the abstract is not understandable as vaccinated people can acquire the infection and the disease. Infected people can became severely ill and die in the same way as it happens for vaccinated COVID–19–patients.

Author’s response 1: The author fully agrees with the fair comment of the Reviewer. Because the number of words in the Abstract is limited, the following was added to the second paragraph of the Introduction: “People vaccinated by injection of IIV can acquire the influenza infection, but the disease is less severe in those who have been vaccinated. In contrast, LAIV is administered intranasally and directly protects the entry gate of infection, so the risk of infection is significantly lower.”

Point 2: Several recent publications reporting about hospitalized severely ill influenza infected patients do not even mention the vaccination status (Kao et al 2018 Ann Intensive Care, Hussain M et la 2020 Pneumonia).Other found (retrospectively) that vaccinated persons are less often sick after influenza infection but nosocomial infections were also serious in vaccinated patients (Ye LT et al. Int J Env research and Public health 2019;16:1078,Murillo–Zamora E et al. Int J Infectious Dis 2021;112:288–293).

Author’s response 2: Unfortunately, many researchers by default mean influenza vaccination exclusively with an inactivated vaccine. It is unfortunate that the authors you mentioned did not indicate what type of vaccine people were vaccinated with (Liang–Tsai Yeh et al. Int J Environ Res Public Health. 2019; 16(6):1078. doi: 10.3390/ijerph16061078; Efrén Murillo–Zamora et al. Int J Infect Dis. 2021; 112:288–293. doi: 10.1016/j.ijid.2021.09.037) or whether they were vaccinated at all (Mujahid Hussain et al. Pneumonia [Nathan]. 2020; 12:5. doi: 10.1186/s41479–020–00070–7; Kuo–Chin Kao et al. Ann Intensive Care. 2018 Sep 24;8(1):94. doi: 10.1186/s13613–018–0440–4). I'm sure both Taiwan and Mexico used an inactivated vaccine. In contrast, the author of this review considers the live attenuated influenza vaccine, and the conclusions that follow from the results of vaccination with the inactivated vaccine are far from always applicable to it.

Point 3: Chow EJ et al. found that the number of influenza positive patients admitted to the hospital because of respiratory problems who were vaccinated was similar to that non–vaccinated. (JAMANetwork Open 2020 March 30).

Author’s response 3: Despite the fact that the live attenuated influenza vaccine FluMist is widely used in the United States, in the article by Eric J Chow et al (Multicenter Study JAMA Netw Open. 2020; 3(3):e201323. doi: 10.1001/jamanetworkopen.2020.1323) I did not find any mention of the specific vaccine type that the subjects were vaccinated with. I believe that it is impossible to draw any conclusions about the effectiveness or ineffectiveness of influenza vaccination if it is not known what type of vaccine people were vaccinated with.

Point 4: It is necessary to make clear that the virus continue to cause death after more than 100 year after its first appearance.

Author’s response 4: The authors greatly appreciate the Reviewer's completely fair comment. The sentence in Introduction “Like many other infections, influenza is a vaccine–preventable disease” was replaced with the paragraph “The primary measure used today to control infectious diseases is vaccination. Humankind eradicated smallpox and plague, but we are still like children in the face of attacks of influenza virus that continue to cause death after more than 100 year after its first appearance. Since those times, tremendous progress has been made in studying influenza. «The paradox of the current situation is a clear contradiction between the achievements of theoretical science and practical results in the fight against influenza infection. Science has created all theoretical prerequisites for defeating influenza. Nevertheless, the study on influenza and influenza viruses is still at its beginning…» [1] and vaccination remains the main tool of influenza control.”

Point 5: The virus does not reach the blood stream. How should the antibody work to avoid further worsening of the clinical conditions (e.g.hypoalbuminemia,kidney insufficiency etc.). These are crucial questions.

Author’s response 5: Unfortunately, sometimes it does. In 1980s, our team isolated the influenza virus from the cord blood of newborns whose mothers had the flu. This is indeed a very important key question asked from a clinician's point of view. The author, on the other hand, approaches the problem from the other side – as a molecular biologist and virologist–developer of vaccine strains, for whom the most important thing is to prepare a strain in time and understand what properties of the wild parental virus interfere with this. Therefore, this review does not address the clinical course of influenza.

Point 6: Authors should also say that the best preventive measure in the hospitals is wearing masks (Ambrosch A et al. J Hospital Infection 2016;94:143–149). Therefore, the first recommendation (also from the WHO) should be that hospital personnel should always wear masks.

Author’s response 6: The author fully agrees with this fair comment of the Reviewer. The effectiveness of wearing masks to protect against respiratory infections has been successfully demonstrated during the COVID–19 pandemic. As practice has shown, along with vaccination, wearing masks played an important role in reducing the spread of SARS–CoV–2. The following paragraph was inserted into the Introduction section: “Speaking of protective measures against influenza, we should not forget about simple hygiene protective equipment, such as masks. WHO recommends the use of medical masks to fight pandemic influenza and as a precautionary measure during the coronavirus pandemic. Today, in the era of COVID–19, we have begun to forget about such a dangerous infection as the seasonal flu, and in vain. WHO, in its recommendations for the prevention of seasonal influenza, gives several steps, the main of which is vaccination, but does not mention such simple but effective measure as wearing masks [13]. Nevertheless, this measure can reduce the risk of disease in families, communities and especially in hospitals [14].”

Round 2

Reviewer 2 Report

Thank you for the kind answers to my criticisms and for the changes made.

I think that the sentences of the rebuttal should be introduced into the the text.

I mentioned only a few recent papers but there are many more which are critical about the efficacy the influenza vaccines.

Furthermore,we knew for longer time that people can die of nosocomial

influenza infection and it was a real desaster that we did not introduce the

wearing of masks in the hospital during the beginning of the recent pandemic.This should be the first measure.Now we should have enough masks.Even molecular biologists should know this before they learn more about the different forms of vaccines

Author Response

COMMENTS AND SUGGESTIONS FOR AUTHORS

Thank you for the kind answers to my criticisms and for the changes made.

Author’s response: The author greatly appreciates the Reviewer for his kindness and patience.

Point 1: I think that the sentences of the rebuttal should be introduced into the text.

Author’s response 1: The author sincerely thanks the Reviewer for this valuable comment and inserted her considerations expressed in responses 2 and 3 (lines 86–91 and 97–106 of the re–revised manuscript).

 Point 2: I mentioned only a few recent papers but there are many more which are critical about the efficacy the influenza vaccines.

Author’s response 2: Yes, the respected Reviewer is absolutely right, there are a number of such publications, but they usually describe insufficient efficacy of inactivated influenza vaccines. In part, this may be due to the mismatch between the circulating and vaccine strains, especially when it comes to an inactivated vaccine. It is also possible that the not entirely satisfactory quality of some influenza vaccines is precisely due to the biological properties of the parental viruses that are described in this review article. I deeply believe that the quality and effectiveness of any influenza vaccine is largely determined by the right choice of the vaccine strain. Thus, if in the preparation of any vaccines, both inactivated and live, vaccine developer will be guided not only by the antigenic novelty of the strain recommended by WHO, but also will carefully select among similar viruses one that, in addition to antigenic novelty, has all the necessary biological properties described in this article, the effectiveness of influenza vaccines may increased. The stated consideration was added (lines 97–106 of the re–revised manuscript).

Point 3: Furthermore, we knew for longer time that people can die of nosocomial influenza infection and it was a real disaster that we did not introduce the wearing of masks in the hospital during the beginning of the recent pandemic. This should be the first measure. Now we should have enough masks. Even molecular biologists should know this before they learn more about the different forms of vaccines.

Author’s response 3: I fully agree with the position of the Reviewer. If vaccines are not effective enough or they are not available at all, individual hygiene and protection measures must be put forward in the first place. Unfortunately, at the beginning of the recent pandemic, humanity has made many irreparable mistakes. Perhaps, in the case of a wider use of masks, many fatal outcomes could have been avoided. The stated consideration was added (lines 86–91 of the re–revised manuscript).

Round 3

Reviewer 2 Report

I appreciate the statments introduced into the text.

I however still have problems with the first sentence of the abstract.

I would suggest the following sentence (which may be too long):"Influenza is still a frequent seasonal infection of the upper respiratory tract,which may have deadly consequences.This, In spite of the availability of vaccines suggested for persons aove 65 years of age".

The second point is the mention of the fact that the influenza virus does not reach the blood stream (as it is the case for SARS-CoV-2).This sentence can be introduced under point 7 and shortly discussed.

The third point is still the lack of the immunological explanation for the kack of effectivity of the current vaccines because of the antigenic changes of the new variants wich should escape the induced immunity by the vaccines.

It should be made clear that it is worth to continue to invest in producing vaccines against a well known RNA-Virus

Author Response

Response to Reviewer #2 Comments

The third round of revision

COMMENTS AND SUGGESTIONS FOR AUTHORS

I appreciate the statements introduced into the text.

Author’s response 1: The author sincerely thanks the Reviewer for these comments.

Point 1: I however still have problems with the first sentence of the abstract. I would suggest the following sentence (which may be too long):"Influenza is still a frequent seasonal infection of the upper respiratory tract, which may have deadly consequences. This, In spite of the availability of vaccines suggested for persons above 65 years of age".

Author’s response 1: The author sincerely thanks the Reviewer for this valuable comment. The first sentence of the Abstract “Influenza, like many other infectious diseases, is a vaccine-preventable illness” was replaced with “Influenza is still a frequent seasonal infection of the upper respiratory tract, which may have deadly consequences especially for elderly. This, in spite of the availability of vaccines suggested for persons above 65 years of age” as the respectful Reviewer suggested (Lines 7-9 of the R3 version of the MS).

Point 2: The second point is the mention of the fact that the influenza virus does not reach the blood stream (as it is the case for SARS-CoV-2).This sentence can be introduced under point 7 and shortly discussed.

Author’s response 2: The author sincerely thanks the Reviewer for this valuable comment. However, I am not sure that I understand what the respectful Reviewer means. During the first round of reviewing, the Reviewer said that the virus does not reach the blood stream. I said that ‘unfortunately, sometimes it does.’ Viremia is a fairly common symptom of influenza, confirmed experimentally (10.1016/s0882-4010(95)90290-2; 10.1111/j.1537-2995.2007.01264.x). In any case, whether wild flu virus causes viremia or not, I do not fully understand how this relates to the main purpose of this review article, which describes how to select optimal parental viruses for the preparation of influenza vaccines, in particular, LAIV.

Point 3: The third point is still the lack of the immunological explanation for the lack of effectivity of the current vaccines because of the antigenic changes of the new variants which should escape the induced immunity by the vaccines.

Author’s response 3: The author sincerely thanks the Reviewer for this criticism. A large body of literature has been devoted to explanations for the lack of effectivity of the current vaccines because of the antigenic changes of the new variants and escaping from immunity. The author inserted a few sentences regarding antigenic changes and escape immunity to the Introduction section (Lines 29-32 of the R3 version of the MS). With all due respect to the opinion of the Reviewer, I believe that more detailed answers to such questions overload a review article devoted to completely different issues.

Point 4: It should be made clear that it is worth to continue to invest in producing vaccines against a well known RNA-Virus

Author’s response 4: The author sincerely thanks the Reviewer for this valuable comment. The author is deeply convinced that it is worth continuing to invest in the production of influenza vaccines. From one hand, the virus could be called ‘well-known’ but from the other hand for 100 years we still know just a little about this virus which is responsible for 3-5 million severe illnesses worldwide and up to 650 thousand respiratory deaths. This statement was added to the Introduction section (Lines 27-29 of the R3 version of the MS). The author believe that if developers of influenza vaccines will be guided not only by the antigenic novelty of the strain recommended by WHO, but also will carefully select among similar viruses one that, in addition to antigenic novelty, has all the necessary biological properties, the effectiveness of influenza vaccines may increase.
